# OpenReview forum: "FUSE: Full‑spectrum Unlearnable Examples via Spectral Equalization"
_ICML.cc/2026/Conference — ICML 2026 regular_

### Official Review · Reviewer_NCfU · 2026-03-10

**Soundness:** 3
**Presentation:** 3
**Significance:** 3
**Originality:** 3
**Overall Recommendation:** 4
**Confidence:** 2

**Summary:**

The paper introduces Full-spectrum Unlearnable Examples via Spectral Equalization (FUSE), a method that protects data privacy by generating perturbations that make sensitive images unusable for machine learning models while remaining visually imperceptible. Existing methods fail when low-pass filters suppress high-frequency components, allowing models to recover information. FUSE solves this by creating perturbations across the entire frequency spectrum, maintaining unlearnability even under filtering. It uses two strategies: Random Spectral Masking (RSM) to spread perturbations across the spectrum, and Cross-Band Guidance (CBG) to ensure consistency between high- and low-frequency components, preserving semantic integrity.

**Compliance With Llm Reviewing Policy:**

Affirmed.

**Key Questions For Authors:**

1. Have you tested FUSE against attacks that explicitly target the full-spectrum perturbation strategy?
2. Why is rc=0.5 optimal across datasets any theoretical justification beyond empirical stability?
3. Add a brief discussion of theoretical intuitions or bounds for spectrum-agnostic unlearnability.

**Limitations:**

Yes, the authors adequately discuss the limitations of their work.

**Strengths And Weaknesses:**

Strengths
1. The methodology, especially the integration of RSM and CBG, is clearly explained with strong theoretical underpinnings and solid experimental support.
2. RSM + CBG components are complementary and well-motivated.
3. Comprehensive evaluation across multi-datasets, multi-architectures, multiple filtering scenarios, and defenses.


Weaknesses
1. Lacks formal guarantees on why spectrum-agnostic perturbations generalize better.
2. Doesn't deeply explore why/when FUSE might fail (adaptive filtering, extreme cutoffs).
3. rc=0.5 and λ=0.5 work well but may require tuning for new domains.

---

> ### Author Rebuttal · Authors · 2026-03-31
>
> We appreciate your detailed feedback. Below, we address the concerns raised in the weaknesses and questions section. We are happy to provide additional clarification if needed.
> ## **W1 & Q3**
>
> We sincerely thank the reviewer for raising this insightful point. Providing strict formal generalization bounds for unlearnable perturbations in deep neural networks remains an open and highly challenging problem. To the best of our knowledge,  no existing work has addressed this issue. However, we can develop **a theoretical intuition and formalize our reasoning from the perspective of Spectral Bias.**
>
> Specifically, we could leverage the theoretical analyses of spectral bias [1, 2], which indicate that neural networks learn different frequency components unevenly, often exhibiting a strong bias towards lower-frequency or lower-complexity functions, while higher-frequency components require more delicate parameter tuning. Formally, if an unlearnable perturbation relies exclusively on a narrow frequency band, its effectiveness is highly sensitive to the network's spectral bias and filtering operations. From this perspective, our spectrum-agnostic approach encourages perturbations to remain effective under varying spectral masks. By doing so, it forces the noise to distribute its protective properties across multiple frequency bands, fundamentally reducing the over-reliance on any single band. We will add a more detailed discussion regarding this point in our revised paper.
>
> [1] On the Spectral Bias of Neural Networks.
>
> [2] Towards Understanding the Spectral Bias of Deep Learning.
> ## **W2 & Q1**
>
> Beyond the JPEG evaluation (Table 6) and the results under various low-pass (Figures 3 and 5) and high-pass (Table 18) cutoffs already included in the paper, we conducted two additional analyses to address this concern more directly. First, against **[three representative UE defenses](https://anonymous.4open.science/r/icml-5BE2/UE%20defense.png)** (D-VAE [1], AN-SDA [2], and ORTHO PROJ [3]), FUSE consistently **maintains strong unlearnability** across all these settings, substantially outperforming prior methods. While no prior defense explicitly targets full-spectrum perturbations, these representative UE defenses still provide a relevant stress test for FUSE. Second, under **band-pass filtering**, FUSE remains **consistently near random-guess accuracy** across all tested cutoffs, while prior UE methods show stronger dependence on the preserved high-frequency range (see **Reviewer qFk9, Q2** for detailed results). Together, these results provide additional evidence that FUSE remains effective under both **stronger UE-oriented defenses** and **broader spectral transformations**.
>
> [1] Purify unlearnable examples via rate-constrained variational autoencoders.
>
> [2] Detection and defense of unlearnable examples.
>
> [3] What Can We Learn from Unlearnable Datasets?
> ## **W3 and Q2**
>
> We thank the reviewer for this question. A useful intuition is that $r_c$ controls **the balance between the low- and high-frequency branches**. If $r_c$ is too small, the low-frequency branch becomes too narrow to preserve sufficient semantic structure, while if $r_c$ is too large, the high-frequency branch becomes too limited to provide flexible unlearnable signals. A mid-range split is therefore a natural choice, as it allows both branches to **remain informative** and enables effective interaction through CBG. This intuition is consistent with our empirical results. In the appendix Section D.6, we conduct an ablation study of $r_c$ under a low-pass filter on four datasets with different resolutions and characteristics. The best-performing $r_c$ consistently lies near 0.5 across all datasets, and **the performance variation within this range is small**, suggesting that the choice is stable rather than highly dataset-specific.
>
> To further verify this, we applied Bayesian Optimization to automatically search for the best $r_c$ on each dataset (Table 16). The identified values consistently lie near 0.5. A more finely tuned $r_c$ can yield slightly better results, and this search process naturally benefits from the theoretical foundation of Bayesian Optimization.
>
> For $\lambda$, our current experiments (Figure 4) under both unfiltered and low-pass settings show that moderate values perform best. To further verify this,  we conducted an additional ablation experiment on [CIFAR-100](https://anonymous.4open.science/r/icml-5BE2/ablation_lamb_100.pdf) and [SVHN](https://anonymous.4open.science/r/icml-5BE2/ablation_lamb_svhn.pdf). All results consistently show that **moderate values around $\lambda=0.5$ perform best or near best**, while very small or very large values lead to worse unlearnability. This suggests that $\lambda = 0.5$ is a robust default that balances semantic guidance and structural alignment across datasets.
>
> [1] Gaussian Process Optimization in the Bandit Setting: No Regret and Experimental Design.

---

> > ### Author Rebuttal · Reviewer_NCfU · 2026-04-01
> >
> > I thank the authors for their comprehensive and well-structured response. Overall, I find that your rebuttal addresses my primary concerns, and I support the paper’s acceptance.

---

> > > ### Author Response · Authors · 2026-04-02
> > >
> > > We sincerely thank Reviewer NCfU for the recognition of our work and for the positive response. We are glad to see that your earlier concerns have been addressed. Moreover, we will incorporate these experimental results and discussions into the revised version. Thank you once again for your time and for considering our work.

---

### Official Review · Reviewer_k4rB · 2026-03-12

**Soundness:** 3
**Presentation:** 3
**Significance:** 2
**Originality:** 2
**Overall Recommendation:** 4
**Confidence:** 3

**Summary:**

This paper studies the vulnerability of existing unlearnable examples (UEs) under spectral filtering and observes that many current UE methods rely heavily on high-frequency perturbations, which can be removed by simple low-pass filtering. To address this limitation, the authors propose FUSE (Full-spectrum Unlearnable Examples via Spectral Equalization), a method designed to generate spectrum-agnostic perturbations. The approach introduces two key components: Random Spectral Masking (RSM), which randomly removes frequency bands during training to encourage robustness across the spectrum, and Cross-Band Guidance (CBG), which enforces consistency between low- and high-frequency components. Experiments on several datasets and architectures show improved robustness of the generated UEs against spectral filtering operations.

**Compliance With Llm Reviewing Policy:**

Affirmed.

**Final Justification:**

My concerns are all well solved during the rebuttal. Thereby, I raise my score and recommend to accept this work.

**Key Questions For Authors:**

1. Ambiguity in the formulation of loss functions (Eq. 14–15). The inputs to the loss functions in Equations (14) and (15) appear inconsistent. In Eq. (14), the loss takes the model output and ground truth as inputs, while in Eq. (15) the formulation suggests different inputs. This discrepancy easily makes readers confused about understanding the proposed method.

2. Lack of discussion for multimodal generation tasks. Multimodal generative models (e.g., text-to-image or vision-language models) are increasingly important and frequently trained on large-scale image datasets. Since the goal of UEs is to protect data from unauthorized training, it would be valuable to discuss how the proposed method could be applied in multimodal model training scenarios.

3. Limited comparison with stronger purification methods. The experiments only evaluate purification strategies such as Adversarial Training (AT), filtering, and JPEG compression, while the latter two are both closely related to frequency filtering. Since the proposed method explicitly targets frequency-based purification, this comparison may not be sufficiently fair. Stronger or more diverse purification approaches should be considered.

4. Limited novelty and general applicability. The core idea of the method mainly focuses on improving robustness against frequency filtering purification by introducing random spectral masking and cross-band constraints. While effective in this specific setting, the broader applicability and conceptual novelty appear limited. The key mechanism—randomly masking spectral blocks to encourage robustness—does not provide deeper insights into the fundamental properties of unlearnable examples.

**Limitations:**

While the paper addresses an interesting problem and proposes a reasonable solution, the evaluation appears somewhat biased toward frequency-based purification methods, which are precisely the attacks the method is designed to resist. In addition, the novelty and broader insights of the approach are somewhat limited. Strengthening the experimental comparisons and expanding the discussion to broader scenarios (e.g., multimodal training pipelines) would improve the paper.

**Strengths And Weaknesses:**

1. Clear motivation. The paper identifies an interesting limitation of existing unlearnable examples, namely their dependence on high-frequency perturbations and vulnerability to spectral filtering.

2. Conceptually simple design. The proposed Random Spectral Masking and Cross-Band Guidance mechanisms are intuitive and easy to integrate into UE generation pipelines.

3. Empirical validation across datasets and architectures. The paper evaluates the approach on multiple datasets and model architectures, which provides some evidence for the effectiveness of the proposed method.

4. The paper is well-structured and easy to follow.

---

> ### Author Rebuttal · Authors · 2026-03-31
>
> We thank the reviewer for the constructive feedback and for highlighting both the strengths and open questions in our work. Below, we show our response to your concerns in the weaknesses section. We are happy to provide additional clarification if needed.
> ## **Q1**
> We thank the reviewer for pointing this out. We would like to clarify that our intended meaning is that $\mathcal{L}\_{\mathrm{full}}$ is optimized with respect to the perturbation generator $\mathcal{G}\_\psi$, based on the clean sample $(\boldsymbol{x}, y)$, the current classifier $f\_{\theta}$, and the generated perturbation $\boldsymbol{\delta} = \mathcal{G}\_\psi (\boldsymbol{x})$. We will  revise Eq. (14) and the surrounding text to make the training dependencies explicit and consistent with Algorithm 1.
> $$
> \begin{aligned}
> \arg\min\_{\theta}\& \mathbb{E}\_{(\boldsymbol{x}, y) \sim \mathcal{D}\_{c}} \left[ \min\_{\psi}\mathcal{L}\_{\mathrm{full}}\big( \boldsymbol{x}, y; f_\theta, \mathcal{G}\_\psi \big) \right] \\\\
> \text{s.t.}\ & \||\boldsymbol{\delta}\||_\infty \leq \epsilon
> \end{aligned}
> $$
>
> ## **Q2**
> We thank the reviewer for raising this important question. Our current work focuses on UE protection for **image classification**, while multimodal generative models [1, 2] are trained on **paired image-text data** and depend on **cross-modal alignment** objectives. Extending FUSE to such settings would therefore require analyzing not only image-side perturbations, but also their interaction with textual conditioning and multimodal supervision, which is beyond the scope of this paper.
>
> Even so, as an initial step toward this direction, we additionally [evaluate FUSE on CLIP image encoder](https://anonymous.4open.science/r/icml-5BE2/CLIP.png) (ResNet50×4) [3], a representative vision-language model. To make this comparison more explicit, we also report the accuracy drop relative to the clean-trained CLIP model in each setting. FUSE shows the best unlearnable performance while prior UE baselines remain much closer to the clean trained model. This suggests that FUSE can affect **vision-language alignment** beyond standard image classification. We agree that testing on multimodal generative models is an important future direction.
>
> [1] High-resolution image synthesis with latent diffusion models.
>
> [2] Scaling Up Visual and Vision-Language Representation Learning With Noisy Text Supervision.
>
> [3] Learning Transferable Visual Models From Natural Language Supervision.
>
> ## **Q3**
> To address this concern, we additionally test FUSE against [three representative defense methods designed for unlearnable examples](https://anonymous.4open.science/r/icml-5BE2/UE%20defense.png), D-VAE [1], AN-SDA [2], and ORTHO PROJ [3], which are substantially different from simple frequency filtering or JPEG-based purification. The new results show that FUSE consistently remains the most effective UE under all three defenses. Specifically, after applying these results, the model accuracy on FUSE-protected data is **significantly lower than other baselines**. These results suggest that the advantage of FUSE is **not limited to frequency-related purification settings**.
>
> [1] Purify unlearnable examples via rate-constrained variational autoencoders.
>
> [2] Detection and defense of unlearnable examples.
>
> [3] What Can We Learn from Unlearnable Datasets?
>
> ## **Q4**
> We understand the reviewer’s concern that random spectral masking may appear heuristic at first glance. First, we would like to note that full-spectrum effectiveness is not introduced as a special-case robustness objective for a specific setting. Rather, the insight of this work is revealing **a fundamental limitation that is suffered by ALL existing UEs:** they tend to collapse to high-frequency-dominant perturbations during optimization. In other words, ALL existing UEs can be purified by simply applying a low-pass filter. Therefore, we argue that **spectrum-agnostic**, as highlighted in this paper, is a key desirable characteristic that should be broadly applied to (but ignored by) all existing UEs. The key mechanisms, RSM and CBG, are designed explicitly tailored to the limitation we identified above.
>
> This broader applicability is also supported by our additional experiments. Beyond frequency filtering, FUSE **remains stronger under representative UE defense methods** (Question 3 response), and it also shows **more reliable cross-dataset transferability** in both the unfiltered and low-pass settings (Tables 10 and 11). Together, these results suggest that reducing spectral over-specialization improves not only filtering robustness, but also the generality and stability of the learned unlearnable perturbations.

---

> > ### Author Rebuttal · Reviewer_k4rB · 2026-04-01
> >
> > Thanks for the author's detailed response. However, some concerns remain;
> >
> > For Q2, could you clarify how you can transfer the proposed method to a more general task setting?
> >
> > For Q3, could you please show the detailed results?

---

> > > ### Author Response · Authors · 2026-04-02
> > >
> > > We thank the reviewer k4rB for this helpful follow-up. As an initial step beyond standard image classification, we evaluate FUSE with a pretrained **CLIP (ResNet50×4)** model on CIFAR-10 by **freezing the text encoder and most of the image encoder**, while fine-tuning only the **last visual block and the attention pooling layer**. In the current paper, the spectral-invariant objective is instantiated with standard classification supervision, e.g., Eq. (2). In the CLIP setting, this can be naturally adapted by replacing the classifier-based cross-entropy term with a **CLIP-style image-text alignment loss**.
> > >
> > > Concretely, relative to Eq. (2), we keep the same masked perturbed input $T\_{\mathrm{spec}}^{(k)}(x+\delta)$ and replace only the supervision term. The CLIP-style spectral-invariant objective becomes
> > >
> > > $ \mathcal{L}\_{\text{inv-spec}}^{\text{CLIP}} = \mathcal{L}\_{\text{con}}(S) + \mathcal{L}\_{\text{con}}(S^\top), \quad S\_{i,j} = \tau \langle V_i, T_j \rangle, $
> > >
> > > where $\mathcal{L}\_{\mathrm{con}}(S)$ and $\mathcal{L}\_{\mathrm{con}}(S^\top)$ represent the image-to-text ($\mathcal{L}\_{i \to t}$) and text-to-image ($\mathcal{L}\_{t \to i}$) contrastive losses, respectively. Here, $V_i = f\_{\theta}^{\mathrm{img}}(T\_{\mathrm{spec}}^{(k)}(x_i+\delta_i))$ represents the perturbed image embedding, $T_j = f^{\mathrm{text}}(t_j)$ represents the text embedding from the frozen text encoder, $\tau$ is the temperature parameter, and $\langle \cdot, \cdot \rangle$ denotes the similarity. Thus, the spectral perturbation framework from Eq. (2) remains unchanged, and only the classifier-based supervision is replaced by a CLIP-style image-text contrastive objective.
> > >
> > > This direction is also motivated by prior evidence. Work on **spectral bias** suggests that neural networks learn different frequency components unevenly, often favoring lower-frequency or lower-complexity functions, while recent studies show that vision-language models can be vulnerable to **frequency-domain perturbations** [1, 2]. More generally, adversarial perturbations designed against large visual foundation models have also been shown to transfer to their downstream models [3].
> > >
> > > From this perspective, a perturbation that remains effective in a more **spectrum-agnostic** manner, rather than relying heavily on a narrow frequency band, may also remain effective more consistently in broader image-based multimodal training settings. In terms of methodology, the most directly transferable part of FUSE is its **spectral mechanism**. In the current paper, the supervision is instantiated as standard classification supervision; in a more general task setting, this part can in principle be replaced by the corresponding **task-specific training signal**, while keeping the same spectrum-agnostic perturbation design. We therefore view our CLIP experiment as **an initial proof-of-concept** in this direction.
> > >
> > > We apologize for the inconvenience of using anonymous links in our previous response Q2 and Q3 due to character limits. We now explicitly report the detailed Q3 numerical results below. As shown in the table, FUSE comprehensively outperforms existing UEs against these advanced, non-frequency-based purification defenses. Unlike prior UEs that heavily rely on specific frequency bands, FUSE's spectrum-agnostic nature allows it to evade these diverse purifications.
> > >
> > > Moreover, we will incorporate these experimental results and discussions into the revised version and we hope these additional results effectively address the reviewer's concerns.
> > >
> > > | Method |   D-VAE   |  AN-SDA   | ORTHO PROJ |
> > > | :----: | :-------: | :-------: | :--------: |
> > > | CLEAN  |   93.29   |   92.76   |   90.16    |
> > > |  EMN   |   91.42   |   88.01   |   65.17    |
> > > |  LSP   |   91.20   |   64.34   |   87.99    |
> > > |   AR   |   91.77   |   80.20   |   13.03    |
> > > |  OPS   |   88.95   |   78.83   |   87.94    |
> > > |  FUSE  | **12.23** | **25.07** | **10.50** |
> > >
> > > [1] On the Reliability of Vision-Language Models Under Adversarial Frequency-Domain Perturbations.
> > >
> > > [2] SPARTA: Spectral Prompt Agnostic Adversarial Attack on Medical Vision-Language Models.
> > >
> > > [3] Transferable Adversarial Attacks on SAM and Its Downstream Models.

---

### Official Review · Reviewer_1KCo · 2026-03-12

**Soundness:** 2
**Presentation:** 2
**Significance:** 2
**Originality:** 3
**Overall Recommendation:** 4
**Confidence:** 4

**Summary:**

This paper address the limitation of existing UE methods, which exhibit a critical failure once low-pass filtering is applied. The authors propose FUSE, a framework that distributes perturbations across the entire frequency spectrum rather than concentrating them solely in high-frequency components. According to the paper, the resulting perturbations preserve visual fidelity while remaining unlearnable under a range of filtering operations and training strategies. Experimental results further indicate that FUSE achieves more stable and robust unlearnability than prior methods, particularly under low-pass filtering and data augmentation defenses.

**Compliance With Llm Reviewing Policy:**

Affirmed.

**Final Justification:**

All the concerns a well-resolved. We believe FUSE is a useful tool and a meaningful advancement over previous methods, and look forward to deeper theoretical investigations concerning the interaction between UE mechanisms, model architecture, and training protocol.

**Key Questions For Authors:**

- can the FUSE perturbations constructed using ResNet-18 generalize to models beyond CNNs and ViT?
- are the generated FUSE perturbations loss-sensitive?
- can a normal model trained with clean data extract meaningful information from perturbed FUSE image?
- how to ensure the perturbed image is nearly visually unchanged to humans?

**Limitations:**

Yes

**Strengths And Weaknesses:**

Strengths:
- Identify a critical vulnerability of existing UEs and propose a systematic method to generate full-spectrum perturbations.
- The paper is well-written and easy to follow.

Weakness:
- One of my major concern is that the cross-architecture test is limited. This is crucial as in practice one can not predict which model architecture will be used in subsequent training. In table 2, most models are CNNs, only ViT is transformer. As the surrogate model (ResNet-18) is also CNN, the results may overestimate the cross-architecture performance. Moreover, I find the result of ViT is not convincing. The performance in table 1 (i.e., with ViT itself as the surrogate model) of ViT when trained using clean data is only 66.16, indicating that the current setting is insufficient for training a descent ViT (maybe due to the lack of enough training data). In cases, where the ViT can be properly trained, it may achieve higher accuracy when trained using FUSE data.
- The ablation study is not sufficient.
    - the effect of training loss is not discussed. How does training loss, especially when different from that used for FUSE perturbation construction, affect subsequent model performance?
    - the ablation for CBG do not reflect the contribution of components within. Is matching classifier logits or structural similarity more important?
- Limited evaluation on small datasets. The experiments are mainly conducted on relatively small datasets such as CIFAR-10, CIFAR-100, and SVHN. It remains unclear whether the proposed method can maintain similar effectiveness on larger and more complex datasets such as ImageNet.

---

> ### Author Rebuttal · Authors · 2026-03-30
>
> We greatly appreciate your constructive and detailed feedback! Below are our responses to your questions and concerns in the weaknesses part. We are happy to provide additional clarification if needed.
>
> ## **W1 & Q1**
>
> To directly address this concern, we conducted two additional experiments.
>
> First, using ResNet-18 as the surrogate model, we evaluated [transferability to three more architecture](https://anonymous.4open.science/r/icml-5BE2/Cross%20Architecture.png): ConvNeXt, MLP-Mixer, and Swin-T. FUSE achieves **the lowest downstream accuracies** on all three targets, outperforming existing UE baselines. This substantially broadens the architectural coverage beyond the original CNN-heavy setting and shows that FUSE is not tied to standard CNN-style targets.
>
> Second, to directly address the reviewer’s concern about ViT training adequacy, we [retrained the ViT with a much longer schedule](https://anonymous.4open.science/r/icml-5BE2/ViT%20extended%20training.png) (**300 epochs instead of 50 epochs**). The results show that FUSE still performs best, confirming that its effectiveness is not an artifact of insufficient ViT training.
>
> ## **W2a & Q2**
>
> Based on Weakness 2(a), our understanding is that the reviewer is asking whether FUSE remains effective when the subsequent model is trained with a loss different from that used for perturbation construction. To directly examine this loss-mismatch setting, we conducted [additional experiments with different downstream training losses](https://anonymous.4open.science/r/icml-5BE2/Train%20loss.png). FUSE consistently remains **the strongest method across these settings**, indicating that its effectiveness does not rely on a specific loss used in subsequent training. This suggests that FUSE is robust to such loss mismatch rather than narrowly tied to a particular training objective. If the reviewer intended a broader notion of loss sensitivity, we would be happy to further clarify.
>
> ## **W2b**
>
> We thank the reviewer for this suggestion. To better disentangle the internal roles of CBG, we further conducted a component-wise ablation. The results show that the **semantic guidance contributes more directly to the unlearnability gain than using structural similarity alone**, while **full CBG still performs best**, indicating that the two terms are complementary rather than redundant. (see **Reviewer Pyeo, W2** for detailed results)
>
> The structural-similarity term mainly **regularizes cross-band interaction and preserves perceptual plausibility**. This is also supported by Appendix D.9 (Table 19), where removing the structural-similarity term degrades PSNR/SSIM and worsens LPIPS/MSE, indicating reduced visual fidelity and structural preservation.
>
> ## **W3**
>
> Thanks for this constructive comment. In fact, we have already evaluated FUSE on the larger and more complex ImageNet* setting under varying low-pass cutoff frequencies in Appendix D.1, Fig. 5(b). The same trend remains consistent: FUSE persistently yields **the lowest downstream accuracy across different cutoff values**. This provides additional evidence that the effectiveness of FUSE is not limited to small datasets and remains consistent on a more challenging large-scale ImageNet-based setting.
>
> ## **Q3**
>
> Again, thank you for this constructive comment. In the UE setting, the key issue is **whether the protected images still provide useful training signals**. We have already evaluated how the proportion of perturbed training samples affects unlearnability in Appendix D.4 (Tables 12 and 13), which answers the reviewer's question. Specifically, this analysis quantifies how much additional information a model can acquire from UEs by comparing performance between training on clean-only data and training on mixed clean-and-UE data. The results show that, under both low-pass and unfiltered settings, **FUSE remains closest to the clean-only baseline across most ratios**, suggesting that a standard model is able to extract only minimal meaningful information from FUSE-perturbed images.
>
> ## **Q4**
>
> We thank the reviewer for this question. The visual fidelity of perturbed images depends on the perturbation budget $\epsilon$. Following common practice in prior UE works, we set $\epsilon = 8/255$, which provides a good balance between imperceptibility and unlearnability. This is supported by our appendix analysis. Fig. 6 shows that perturbations remain nearly imperceptible at small budgets, while Table 9 shows that $\epsilon = 8/255$ already achieves strong unlearnability under both unfiltered and low-pass settings. In addition, Figure 9 shows that FUSE images remain visually similar to clean samples across different low-pass cutoffs, and the [additional visual-quality results](https://anonymous.4open.science/r/icml-5BE2/Visual%20Quality.png) further show that FUSE does not compromise imperceptibility compared with existing UE methods.

---

> > ### Author Rebuttal · Reviewer_1KCo · 2026-04-04
> >
> > We thank the authors for their clear and detailed response. Most of major concerns are well addressed. We still have some questions about the non-matching training loss results. As far as we understand, UE methods create abnormal samples that break the gradient-based updates. Thus, it should be training protocol-specific. On the other hand, since human can recognize the perturbed images, the semantic information is largely retained after the perturbation. So there must be ways to extract those information.
> >
> > In this sense, the practical question is that, under the **common family of training protocols** (of cause not all possible protocols), does FUSE constantly prevents the model from learning information from the perturbed samples? Based on the above, we think the current results of loss ablation are not sufficient. Can the authors provide more details (e.g., hyper-parameter settings) about the loss ablation experiments beyond just loss names? In addition, can the authors explore more boarder types of training settings, e.g., with different levels of regularization? If the authors can provide more elaborated clarifications to those questions, we would like to increase the score.

---

> > > ### Author Response · Authors · 2026-04-05
> > >
> > > We thank the reviewer for the constructive feedback. To directly address the reviewer’s question, we have conducted extensive additional experiments and discussions. Our additional results suggest that the effectiveness of FUSE **is not narrowly tied to any single training setting**.
> > >
> > > To clarify our loss-ablation setting: the **FUSE perturbation construction is kept exactly the same as in the main paper**, and only the **subsequent model training loss** is changed. To ensure reproducibility and address the query regarding experimental details, we provide the specific hyperparameters used in our loss-mismatch experiments:
> > >
> > > *   **Focal Loss:** Focusing parameter $\gamma = 2.0$, and class balancing factor $\alpha = 1.0$.
> > > *   **Label Smoothing:** Smoothing factor $\sigma = 0.1$.
> > > *   **Training Protocol:** Backbone=ResNet-18, Optimizer=SGD (momentum=0.9), Learning Rate=0.1, Batch Size=512, Total Epochs=50.
> > > *   **Regularization:** Weight Decay = $5 \times 10^{-4}$.
> > >
> > > To further address the reviewer’s question about broader training settings, we expanded the evaluation from a single regularization ablation to a broader set of representative training protocols. Specifically, beyond the loss-mismatch experiments, we tested sensitivity to multiple optimization-related factors, including optimizer family, learning rate, scheduler choice, momentum, Adam-style beta parameters, and weight decay. As shown in three supplementary [$\color{red}\textbf{tables}$](https://anonymous.4open.science/r/icml-5BE2/training%20protocols.pdf) provided in the anonymous link, across this broad set of training protocols, FUSE remains consistently effective, suggesting that its unlearnability is not narrowly tied to a single default optimization setup.
> > >
> > > Moreover, we evaluated downstream protocols that explicitly enhance semantic information extraction, including **adversarial training** (AT) and **augmentations** such as Mixup/CutMix (Table 5). AT encourages the model to rely on more robust, semantically meaningful representations [1], while Mixup/CutMix improve generalization and reduce reliance on local or fragile patterns [2, 3]. Under all these settings, FUSE remains highly effective, suggesting that these protocol changes do not substantially improve the model’s ability to recover useful information from FUSE-perturbed samples.
> > >
> > > Overall, these results combined with those in the main paper indicate that FUSE remains consistently effective across a broad family of practical training settings, spanning optimization hyperparameters, training dynamics, input semantic enhancement protocols, different training objectives, and cross-architecture settings (Table 2).
> > >
> > > In terms of the possibility of semantic information extraction from UEs, we note that this concern is in fact closely aligned with the motivation of our method. On the one hand, prior literature suggests that deep networks often **rely on shortcut or easy-to-learn predictive cues**, rather than necessarily prioritizing human-salient semantics [4, 5, 6], which also helps explain why existing UE methods can successfully hinder learning semantic information [7]. On the other hand, our work further shows that prior UE methods do not explicitly ensure the stability of such misleading cues across spectral conditions. From the spectral perspective, by using low-pass filtering as an input-side training protocol, we find that prior UE methods become much more **vulnerable once their high-frequency cues are suppressed**, since the subsequent model can then more easily extract useful semantic information from the protected samples. This observation directly motivates FUSE. Rather than relying on a fragile high-frequency shortcut, FUSE adopts a **spectrum-agnostic design** that encourages the perturbation to remain effective across spectral bands, making it substantially more robust under such settings. The above experimental results validate the effectiveness of FUSE under this broader family of training settings. We also agree that a more systematic characterization of the interaction between UE mechanisms, model architecture, and training protocol is an important future direction.
> > >
> > > We will incorporate these results and discussions into the revision, hoping they effectively address your concerns.
> > >
> > > [1] Towards Deep Learning Models Resistant to Adversarial Attacks
> > >
> > > [2] mixup: BEYOND EMPIRICAL RISK MINIMIZATION
> > >
> > > [3] CutMix: Regularization Strategy to Train Strong Classifiers with Localizable Features
> > >
> > > [4] ImageNet-trained CNNs are biased towards texture.
> > >
> > > [5] Shortcut learning in deep neural networks.
> > >
> > > [6] Which Shortcut Cues Will DNNs Choose?
> > >
> > > [7] Unlearnable Examples: Making Personal Data Unexploitable.

---

### Official Review · Reviewer_qFk9 · 2026-03-12

**Soundness:** 2
**Presentation:** 3
**Significance:** 3
**Originality:** 3
**Overall Recommendation:** 4
**Confidence:** 4

**Summary:**

The paper exposes a critical vulnerability in current Unlearnable Examples (UEs): their protective perturbations are primarily concentrated in high-frequency bands, making them easily circumvented by simple low-pass filtering. To address this, the authors propose FUSE, a framework that trains a perturbation generator to produce spectrum-agnostic UEs. FUSE achieves this through two novel components. First, Randomized Spectral Masking (RSM) stochastically drops frequency bands during training and utilizes a spectral entropy regularizer to force the perturbation energy to distribute evenly across the frequency spectrum. Second, Cross-Band Guidance (CBG) enforces mutual consistency between low- and high-frequency components by aligning their structural similarities and semantic classifier logits. Extensive empirical evaluations demonstrate that FUSE successfully maintains unlearnability under various spectral filtering defenses (e.g., low-pass filtering, JPEG compression) while preserving cross-architecture and cross-dataset transferability.

**Compliance With Llm Reviewing Policy:**

Affirmed.

**Key Questions For Authors:**

(1) Could the authors provide a comprehensive quantitative comparison of visual fidelity (using PSNR, SSIM, and LPIPS) against the representative baselines under the same perturbation budget? I strongly recommend including this comparison in the main text to convincingly demonstrate that the full-spectrum perturbation does not compromise imperceptibility compared to existing high-frequency-biased methods.

(2) How might FUSE perform against an adaptive attacker who has full knowledge of your spectral equalization strategy? For instance, if an attacker applies a band-pass filter to isolate mid-frequencies, would the unlearnability hold?

(3) Table 7 highlights the training and inference efficiency of FUSE compared to the generator-based GUE. However, how does the end-to-end computational cost of FUSE compare to optimization-based methods (like EMN or LSP) that directly optimize the noise tensor without training a neural generator?

**Limitations:**

Yes

**Strengths And Weaknesses:**

Strength:

The empirical evaluation is highly rigorous. The authors test across multiple datasets (CIFAR-10, CIFAR-100, SVHN, ImageNet), various network architectures, and against a comprehensive suite of defenses including low/high-pass filtering, JPEG compression, data augmentations (Cutout, CutMix), and Adversarial Training.


Weakness:

While the authors evaluate image fidelity metrics (PSNR, SSIM, LPIPS, MSE) in Appendix D.9, this is strictly limited to an ablation study (comparing FUSE with and without the structural similarity term). There is no quantitative comparison of image fidelity between FUSE and the baseline methods (e.g., EMN, LSP, TUE, GUE).



(1) Soundness:
Strengths: The empirical evaluation is highly rigorous. The authors test across multiple datasets (CIFAR-10, CIFAR-100, SVHN, ImageNet), various network architectures, and against a comprehensive suite of defenses including low/high-pass filtering, JPEG compression, data augmentations (Cutout, CutMix), and Adversarial Training.
Weaknesses: While the authors evaluate image fidelity metrics (PSNR, SSIM, LPIPS, MSE) in Appendix D.9, this is strictly limited to an ablation study (comparing FUSE with and without the structural similarity term). There is no quantitative comparison of image fidelity between FUSE and the baseline methods (e.g., EMN, LSP, TUE, GUE).

(2) Presentation:
Strengths: The manuscript is exceptionally well-structured, with a clear and compelling narrative flow from problem identification to methodology and evaluation.
Weaknesses: The notation in the final optimization objective (Eq. 14) is slightly dense and could benefit from a brief expanded explanation.

(3) Significance:
Strengths: The paper addresses a highly relevant and pressing issue: protecting personal data from unauthorized commercial model training.

(4) Originality:
Strengths: While frequency-domain analysis is common in general adversarial robustness , the application of spectral equalization specifically to prevent unlearnable examples from collapsing into narrow spectral regions is a novel perspective.

---

> ### Author Rebuttal · Authors · 2026-03-30
>
> We thank the reviewer for the positive and constructive assessment of our work. Below, we respond to your questions and concerns in the weaknesses section. We are happy to provide additional clarification if needed.
>
> ## **Soundness Weakness & Question 1**
>
> To address this concern, we conducted an additional experiment comparing FUSE with representative UE methods under the same perturbation budget ($\epsilon = 8/255$). We report **[the average image-fidelity metrics over 100 CIFAR-10 samples](https://anonymous.4open.science/r/icml-5BE2/Visual%20Quality.png)**. The results show that **the visual quality of FUSE-perturbed images is comparable to existing UE methods**. FUSE achieves the best PSNR and LPIPS among all compared methods, while remaining highly competitive on SSIM and MSE. These results suggest that the proposed full-spectrum perturbation does not compromise imperceptibility compared with existing UE methods. We will include this comparison in our revised paper.
>
> ## **Presentation Weakness**
>
> We agree that the final objective can be presented more clearly. In our framework, $ \theta$ denotes the surrogate classifier parameters, $\mathcal{G}\_{\psi}$ denotes the perturbation generator, and the perturbation is produced as $\boldsymbol{\delta}=\mathcal{G}\_{\psi}(\boldsymbol{x})$. As shown in Algorithm 1 in the appendix, we alternate between two steps: we first optimize the classifier by CE loss while fixing the generator, and then optimize the generator by $\mathcal{L}\_{\mathrm{full}}$ while fixing the classifier. We will add this brief clarification in the revision.
>
> ## **Q2**
>
> We thank the reviewer for this insightful question. To the best of our knowledge, FUSE is **the first UE method explicitly designed for full-spectrum unlearnability**, and there are currently no prior UE defenses specifically tailored to target this setting. To directly address the reviewer’s example of an adaptive attacker isolating mid-frequencies, we additionally evaluate FUSE under **band-pass filtering** on the full CIFAR-10 dataset. FUSE consistently maintains near random-guess accuracy across all cutoff values. In contrast, the other UE methods show a clear dependence on the preserved frequency range: **their test accuracy generally decreases as the retained band becomes wider and preserves more high-frequency content**, indicating stronger unlearnability when more high-frequency perturbation signals survive. This trend is consistent with our main claim in the paper that prior UE methods rely more heavily on high-frequency components. Overall, these results provide **a more complete performance** of FUSE under broader spectral transformations. These additional results therefore complement our low-pass and high-pass analyses, while further showing that **FUSE remains consistently unlearnable even under broader spectral filtering conditions**.
>
> |Band-pass filter|0.4-0.6|0.3-0.7|0.2-0.8|
> |:-|:-:|:-:|:-:|
> | EMN|20.04|17.45|12.45|
> | LSP|34.40|33.04|32.57|
> | TUE|46.29|22.48|14.96|
> | GUE|32.41|23.50|13.73|
> | PUE|20.20|12.42|11.37|
> | FUSE|**10.24**|**10.46**|**10.90**|
>
> ## **Q3**
>
> We further compare the computational cost of FUSE with representative UE methods using per-epoch training time on CIFAR-10 under the same setting. We note that LSP follows a fundamentally different paradigm from optimization-based or generator-training UE methods. Specifically, instead of iteratively optimizing perturbations, it directly synthesizes linearly separable shortcut perturbations, and the original paper explicitly states that this process does not require solving optimization problems. While LSP may be computationally attractive in terms of perturbation construction, its cost is not directly comparable to per-epoch training time because it does not involve an iterative training process analogous to EMN, GUE, or FUSE. For this reason, we do not include LSP in our per-epoch training-cost table.
>
> The results shown in the table suggest that FUSE introduces only a moderate additional cost over lightweight optimization-based baselines such as EMN, while remaining far more efficient than GUE. Therefore, FUSE **does not introduce much additional training overhead** in practice. This is also consistent with the design of FUSE, which means its extra computation mainly comes from lightweight spectral operations, such as frequency decomposition and randomized spectral masking.
>
> |Method|CIFAR10 Training Time (s/epoch)|
> |:-:|:-:|
> |EMN|**13.51**|
> |TUE|17.78|
> |GUE|321.10|
> |PUE|287.01|
> |FUSE|21.24|

---

> > ### Author Rebuttal · Reviewer_qFk9 · 2026-04-03
> >
> > Authors partially resolved my concerns. I would like to keep weak accept recommendation.

---

> > > ### Author Response · Authors · 2026-04-03
> > >
> > > We sincerely thank Reviewer qFk9 for the recognition of our work and for the positive response. Moreover, we will incorporate these experimental results and discussions into the revised version. Thank you once again for your time and for considering our work.

---

### Official Review · Reviewer_Pyeo · 2026-03-23

**Soundness:** 3
**Presentation:** 3
**Significance:** 3
**Originality:** 3
**Overall Recommendation:** 5
**Confidence:** 3

**Summary:**

Unlearnable Examples (UEs) are specifically perturbed images/representations that are hard to learn during model training to protect privacy-sensitive training data.
The paper points out that existing UEs can be easily circumvented through a low-pass frequency filtering with a carefully designed motivation experiment (Figure 1), which suggests that previous methods only perturbed the high-frequency representations rather than the whole spectrum.
To solve the problem, the authors propose a spectrum-agnostic perturbing method, named Full-spectrum Unlearnable examples via Spectral Equalization (FUSE). FUSE consists of a Random Spectral Masking (RSM) that randomized spectral band suppression during training to simulate diverse distortions. It ensures perturbations to remain robust when certain spectral frequency is absent. Another component is Cross-Band Guidance (CBG), which boosts the trade-off between unlearnability and utility of the UEs. Low-frequency components can inherit unlearnability from high frequencies and high-frequency components can preserve semantic fidelity from low frequencies.
Extensive experiments across multiple datasets, architectures, and spectral filtering demonstrate the strong protection achieved by FUSE.

**Compliance With Llm Reviewing Policy:**

Affirmed.

**Final Justification:**

I believe the new ImageNet results further support the training objective design. Since the authors have addressed all of my concerns, I will raise my score accordingly.

**Key Questions For Authors:**

I have written my questions as same as the weakness I proposed above.

**Strengths And Weaknesses:**

Strengths:

1. The motivation for proposing FUSE is strong and supported by convincing evidence.

2. The experiments under both low-pass filtered and unfiltered settings show that FUSE is intrinsically effective in both cases, maintaining strong unlearnability across the full spectrum.

3. The authors thoroughly decompose and explain the training objectives into smaller components, making the methodology clear and easy to understand.

4. In my view, defining the cut-off frequency for the low-pass filter is important. Under different cut-off settings, FUSE shows consistent improvements, whereas the other baselines exhibit noticeable fluctuations.

Weaknesses:
1. My concern is less about the canonical UE objective itself, and more about the practical evaluation setting. While UE methods are commonly evaluated by training on fully protected datasets and measuring the resulting clean-test accuracy drop, in realistic scenarios only a subset of images may be protected, and the attacker may mix them with clean open-access data. Therefore, what if you only perturb certain classes within a training dataset, then what will the result be like on these "protected" classes against the other classes?
2. Although the authors provide ablation studies on the effectiveness of RSM and CBG, I would also like to see more direct evidence for the roles of the two loss terms. For example, it would be helpful to show how the $\\mathcal{L}\_{\\mathrm{struct}}$ term preserves perceptual plausibility, and how the $\\mathcal{L}\_{\\mathrm{guide}}$ term inherits or transfers unlearnable features from the high-frequency representations.

---

> ### Author Rebuttal · Authors · 2026-03-30
>
> We appreciate the reviewer’s positive and constructive feedback on our work. Below, we respond to your concerns in the weaknesses section. We would be glad to provide further clarification if helpful.
>
> ## **W1**
>
> We thank the reviewer for this important practical question. We agree that, in realistic settings, only a subset of training data may be protected, while the attacker can still mix them with clean open-access data. To address this, we additionally evaluate FUSE under a **partial protection setting** on CIFAR-10, where only **[one (bird)](https://anonymous.4open.science/r/icml-5BE2/classwise_protect_bird.pdf)** or **[multiple](https://anonymous.4open.science/r/icml-5BE2/classwise_protect_multi.pdf)** classes are perturbed, and the remaining classes stay clean. Following the protocol used in prior UE work such as EMN, we analyze the resulting confusion matrices and per-class accuracy.
>
> The results show that the **protected classes become effectively unlearnable**, with their accuracy dropping to nearly 0% (similar to EMN), while the **unprotected classes remain largely unaffected**. Moreover, the confusion is clearly **asymmetric**: protected-class samples are misclassified into other classes, but clean samples are rarely mapped back to the protected classes.
>
> In addition, as already reported in our appendix Section D.4, we have evaluated how much additional information a model can acquire from UEs by comparing performance between training on clean-only data and training on mixed clean-and-UE data. The results show that, under both low-pass and unfiltered settings, **FUSE remains closest to the clean-only baseline across most ratios**, suggesting that a standard model is able to extract only minimal meaningful information from FUSE-perturbed images.
>
> ## **W2**
>
> To better disentangle FUSE internal roles, we further conducted a component-wise ablation. The new results show that **under low-pass filtering, semantic guidance contributes more directly to the unlearnability gain than structural similarity alone**. This is consistent with our design: the guidance term aligns the low-frequency branch with the stronger unlearnable effect induced by the high-frequency branch, thereby encouraging comparable protection across frequency bands, especially under low-pass filtering.
>
> |                               | Low-pass filter | Unfiltered |
> | :---------------------------: | :-------------: | :--------: |
> |            w/o CBG            |      17.00      |   11.50    |
> |   w/ semantic guidance only   |      15.55      |   11.49    |
> | w/ structural similarity only |      16.78      |   11.68    |
> |           Full CBG            |    **10.13**    |  **9.41**  |
>
> By contrast, the structural-similarity term is not the main source of unlearnability by itself. Its role is primarily to regularize the cross-band interaction, stabilize optimization, and preserve the visual quality of generated UEs. This is also supported by Appendix D.9 (Table 19), where **removing** the structural-similarity term **degrades PSNR/SSIM and worsens LPIPS/MSE**, indicating reduced visual fidelity and structural preservation.
>
> Importantly, full CBG performs best in both low-pass and unfiltered settings, which indicates that these two terms are not redundant. Rather, semantic guidance improves cross-band unlearnability transfer, while structural similarity provides complementary regularization that improves coherence, fidelity, and overall effectiveness.

---

> > ### Author Rebuttal · Reviewer_Pyeo · 2026-04-01
> >
> > Thanks to the authors for the comprehensive responses. My concerns regarding the practical scenarios of UE have been fully addressed. I also found the additional results related to Weakness 2 quite insightful, particularly the observation that structural similarity remains relatively high for UEs regardless of whether it directly contributes to their generation or optimization. Since the reported improvements are relatively modest, I would like to ask whether the authors could provide a similar table to those in Weakness 2 and Table 19 on a higher-resolution dataset such as ImageNet-1K. A subset of ImageNet would also be sufficient, as it would further strengthen the evidence for the method’s generalization.

---

> > > ### Author Response · Authors · 2026-04-02
> > >
> > > We thank the reviewer Pyeo for the helpful follow-up suggestion. Following this suggestion, we conducted two additional experiments on ImageNet* (the first 100 classes of ImageNet).
> > >
> > > First, we performed a component-wise ablation similar to the table in Weakness 2. The results show the same overall pattern as on CIFAR-10: **semantic guidance contributes more directly to the unlearnability gain, especially under low-pass filtering**, while **structural similarity alone provides only limited improvement**. At the same time, **full CBG performs best overall**, confirming that the two terms are complementary rather than redundant. These results further support our interpretation that the guidance term is the main source of cross-band unlearnability transfer, whereas the structural term mainly plays a regularizing role.
> > > |                               | Low-pass filter | Unfiltered |
> > > | :---------------------------: | :-------------: | :--------: |
> > > |            w/o CBG            |      12.38       |    3.84    |
> > > |   w/ semantic guidance only   |      6.17       |    1.89    |
> > > | w/ structural similarity only |      11.05      |    3.16    |
> > > |           Full CBG            |    **4.45**     |  **1.10**  |
> > >
> > > Second, we conducted a fidelity comparison similar to Table 19 by evaluating the structural-similarity component on ImageNet*. The results again show a consistent trend: **removing the structural-similarity term degrades visual fidelity**, while the full FUSE objective achieves clearly better perceptual quality.
> > >
> > > |Method|PSNR ↑|SSIM ↑|LPIPS ↓|MSE ↓|
> > > |:-:|:-:|:-:|:-:|:-:|
> > > |w/o structure|   35.26   |   0.9846   |   0.0293   |   0.000314   |
> > > |FUSE|**39.22**|**0.9974**|**0.0123**|**0.000128**|
> > >
> > > For all experiments, we keep the same perturbation budget of $\epsilon = 8/255$ as in the rest of the paper. Under the same bound, higher-resolution natural images can often yield somewhat better fidelity metrics in absolute value, since the same per-pixel perturbation budget tends to produce a smaller relative distortion over the full image. More importantly, our conclusion does not depend on the values themselves: **the same qualitative roles of the two terms remain consistent on higher-resolution data**. Overall, these results further strengthen the evidence that semantic guidance mainly improves cross-band unlearnability transfer, while structural similarity primarily preserves visual fidelity.
> > >
> > > Moreover, we will incorporate these experimental results and discussions into the revised version and we hope these additional results effectively address the reviewer's concerns.

---

### Decision · Program_Chairs · 2026-04-30

**Decision:**

Accept (regular)

**Comment:**

The authors aim to address the limitation of existing UE methods, which exhibit a critical failure once low-pass filtering is applied (an important problem). Then, FUSE is proposed as a framework that distributes perturbations across the entire frequency spectrum rather than concentrating them solely in high-frequency components, opening an interesting research direction.

After two-round discussions, all the concerns are clarified by the authors, and reviewers believe that this paper makes solid contributions to the field.